# Tackling Resistance to Cancer Immunotherapy: What Do We Know?

**DOI:** 10.3390/molecules25184096

**Published:** 2020-09-08

**Authors:** Soehartati A. Gondhowiardjo, Vito Filbert Jayalie, Riyan Apriantoni, Andreas Ronald Barata, Fajar Senoaji, IGAA Jayanthi Wulan Utami, Ferdinand Maubere, Endang Nuryadi, Angela Giselvania

**Affiliations:** 1Faculty of Medicine, Universitas Indonesia, Jakarta 16424, Indonesia; gondhow@gmail.com (S.A.G.); v_lie@hotmail.com (V.F.J.); riyan.apriantoni@gmail.com (R.A.); Andreas16989@gmail.com (A.R.B.); Fajarsenoaji89@gmail.com (F.S.); jayanthiwulan@gmail.com (I.J.W.U.); ferdinandmaubere@gmail.com (F.M.); bob.nuryadi@gmail.com (E.N.); giselvania@gmail.com (A.G.); 2Department of Radiation Oncology, Dr. Cipto Mangunkusumo National General Hospital, Jakarta 10430, Indonesia

**Keywords:** immune escape, cancer, immunotherapy, T cell, immunosuppression, antigen presentation, immune checkpoint, resistance, treatment

## Abstract

Cancer treatment has evolved tremendously in the last few decades. Immunotherapy has been considered to be the forth pillar in cancer treatment in addition to conventional surgery, radiotherapy, and chemotherapy. Though immunotherapy has resulted in impressive response, it is generally limited to a small subset of patients. Understanding the mechanisms of resistance toward cancer immunotherapy may shed new light to counter that resistance. In this review, we highlighted and summarized two major hurdles (recognition and attack) of cancer elimination by the immune system. The mechanisms of failure of some available immunotherapy strategies were also described. Moreover, the significance role of immune compartment for various established cancer treatments were also elucidated in this review. Then, the mechanisms of combinatorial treatment of various conventional cancer treatment with immunotherapy were discussed. Finally, a strategy to improve immune cancer killing by characterizing cancer immune landscape, then devising treatment based on that cancer immune landscape was put forward.

## 1. Introduction

The immune system not only functions to keep foreign pathogens away from the host, it also has a role in suppressing cancer [1]. The precursor of cancerous cells is initially a normal cell that becomes malignant due to multiple mutations within its genome. The mutated genes, when expressed, will result in the production of non-self-antigens. Theoretically, those mutated tumor neo antigens will be recognized and eliminated by our immune system because there has been no tolerance toward those tumor neo antigens. However, cancer cells could develop mechanisms to trick that immune system. 

The mechanism of immune escape is one of the major mechanisms of mutated cells to gain potential to grow and eventually metastasize [2]. Understanding the mechanism of immune escape by cancer cells will provide us with the required insight to potentially develop treatment to reverse that mechanism. In general, immune escape is due to tolerance or failure of two main functions of the immune system: failure of immune recognition and failure of immune attack. Here, we reviewed the process of immune recognition until immune attack toward cancer cells and how cancer cells acquired resistance toward immunotherapy. Furthermore, the importance of various established cancer treatment were discussed followed by a strategy to combine various established treatments with immunotherapy.

## 2. Major Histocompatibility Complex (MHC) Are Core Molecules in Immune Recognition

Generally, there are two main mechanisms for cancer cell recognition by immune cells. The first mechanism is self-presentation of tumor antigen through MHC or human leukocyte antigen (HLA) Class I. The second mechanism is tumor antigen presentation through professional antigen presenting cells (APCs) via MHC or HLA Class II [3]. Both methods of tumor antigen presentation are essential in immune surveillance and recognition [3,4]. Cancer cells generally harbor multiple mutations within their genome. Some parts of the mutated genes may express foreign antigens [5]. These mutated genes are not previously known by the host immune system, therefore, they should be able to be recognized by immune cells [5,6].

The process of antigen presentation of those cancer cells to the immune system requires a properly functioning MHC Class I molecule. MHC Class I molecule is expressed in all nucleated cells as part of the physiologic cellular defense apparatus from foreign antigens. The process of tumor antigen presentation starts with cleavage of tumor antigen in the cytoplasm of the tumor until the tumor antigen becomes a small sequence of amino acids [3,6]. These cleaved segments will combine with MHC Class I molecule in endoplasmic reticulum. These complexes will later be transported to the plasma membrane [6]. The T cells will be able to bind and recognize the tumor antigen-MHC Class I complex on the plasma membrane through a specific T cell receptor.

Many cancers circumvent the process of immune recognition or tumor antigen presentation from the cancer cells to T cells by downregulating MHC Class I within tumor cells [7,8,9,10] (Figure 1a). Various methods were thought to be employed by cancer cells to suppress the expression of MHC Class I. Epigenetic silencing within tumor cells resulted in direct methylation of MHC Class I gene by DNA methyltransferase. The methylation process has been observed in various types of cancers [7,10,11]. The methylation of various essential genes resulted in suppression and downregulation of MHC Class I expression, thus preventing those tumor cells to be recognized by the immune system.

Furthermore, tumor cells can also downregulate MHC Class I molecule by suppressing various MHC Class I transcriptor activators. Loss of Interferon regulatory factor 2 (IRF2) and NOD-like receptor (NLR) family and caspase recruitment (CARD) domain containing 5 (NLRC5) expression have been shown to be correlated with lower expression of MHC Class I and higher expression of immune exhaustion maker [8,9]. These transcriptor activators in cancer cells were suppressed by various means including methylation, copy number loss, or somatic mutation [8,9]. These are some of the known and most common mechanisms of cancers to fail the process of immune recognition via MHC Class I.

Tumor cell recognition through professional antigen presenting cells and the MHC Class II molecule is also an important process in cancer immune surveillance and recognition [12,13]. Though MHC Class II is only expressed in professional antigen presenting cells, its role has been shown to be as critical as MHC Class I in cancer immunotherapy [4]. Tumor cells will continuously proliferate until a point in which the supply of nutrients and oxygen through neo-vasculature becomes inadequate. At that point, the tumor becomes hypoxic and later dies. This process of cell death is commonly immunogenic in nature. This immunogenic tumor death will result in the release of various damage-associated molecular patterns (DAMPs) such as adenosine triphosphate (ATP), high-mobility group protein B1 (HMGB1), and heat-shock proteins that will recruit various antigen presenting cells [14,15].

The immunogenic cell death will also trigger engulfment of tumor debris by phagocytic cells. The phagocytic cells will process the engulfed antigen and finally present it through the MHC Class II molecule to other antigen presenting cells and T helper [16]. The MHC Class II–antigen complex can also be secreted as exosomes, which later fuse with other antigen presenting cells or CD4+ T cells [16]. This fusion will activate that cells and enable those cells to also become potent presenter cells [16]. The antigen presentation toward CD4+ helper T cells through the MHC Class II molecule will enable secretion of various cytokines and chemokines. This antigen presentation of CD4 T cells when in an acute setting will further recruit and activate CD8+ T cells to kill the identified tumor cells.

Nevertheless, tumor antigen presentation through MHC Class II does not always end up with enhanced tumor recognition. When chronic inflammation ensues due to various reasons including chronic necrosis, the chronic exposure of tumor antigens would promote the maturation and recruitment of immunosuppressive immune infiltrate [17] (Figure 1b). Elevated HMBG1 from necrotic cells have been shown to correlate with poor prognosis in many cancer types [18]. Chronic recognition of tumor antigens via antigen presenting cells and MHC Class II molecule does not promote elimination of cancer cells. Instead, the chronic recognition results in local immune-tolerance, thereby promoting tumor growth.

A recent strategy to exploit the tumor recognition via MHC Class II can be performed by inducing MHC Class II expression directly in tumor cells [4]. Though MHC Class II was not commonly expressed in tumor cells, it has been shown to be inducible [4,19,20]. The induction could be carried out by transfecting the tumor cells with the Class II, major histocompatibility complex, transactivator (CIITA) loaded vaccine [4]. This CIITA is a major transactivator of MHC Class II expression. Therefore, by vaccinating the tumor cells with this CIITA loaded vaccine, the tumor cells would express MHC Class II. These tumor cells can then become the antigen presenting cells themselves. This would eventually lead to a strong tumor antigen presentation followed by immune attack. 

## 3. Recruitment and Priming of T Cells

The T cells are essential in establishing and maintaining the homeostasis of immune response. The presence of T lymphocytes within the tumor microenvironment, especially the cytotoxic CD8+ subsets and to some extent the CD4+ subsets, have been shown to correlate with better prognosis of the patient [21,22,23,24,25,26]. The deficiency of CD8+ T cells have been shown in experimental models to result in tumor progression, while tumor regression was observed in the CD8+ proficient model [27]. The spatial distribution of CD8+ T cells within the tumor microenvironment were very variable among different patients and different cancer types [28]. Those distribution of CD8+ T cells were likely related with prognosis, with dense CD8+ T cells between the tumor being more positive than dense tumor stromal CD8+ T cells [29].

The recruitment of T cells to the tumor microenvironment plays an initial important role in determining the magnitude of tumor immune restrain. Generally, T cells from systemic circulation recruited to the tumor microenvironment follows common processes of tethering, rolling, adhesion on endothelial surfaces until diapedesis, as in common inflammatory process [30,31]. The differences in T cell density within the tumor microenvironment was suggested due to differential chemokine expressions in different type of tumors [30,32]. The lack of pro-inflammatory chemokine expression in certain tumors such as CXCL9, CXCL10, and CXCL11 resulted in low CD8+ T cells extravasation to tumoral tissue [32]. Some tumor even expressed chemokines that resulted in the recruitment of T regulatory cells (a potent immunosuppressive T cell) such as CCL22 or T cell repellants such as CXCL12 [30]. All those chemokines over or under expressed by tumor cells are postulated to be responsible for ineffective T cells recruitment, which eventually leads to less inflamed tumors.

Furthermore, vascular endothelial growth factor (VEGF) secreted by tumor cells can also act to impede T cell recruitment [33]. Administration of anti VEGF that somehow normalized the vasculature has been found to significantly increase T cell recruitment into the tumor microenvironment [33]. Once recruited into tumor microenvironment, the T cells need to cross react with tumor antigens through the antigen presentation process. The priming of T cells will enable the T cells to differentiate into effector cytotoxic CD8+ T cells. There are complex processes of T cells priming with the involvement of various cells and molecules. Every defective process could fail the priming process, thus abrogating the T cell attack. Failure of tumor cells to present its antigen through MHC class I to CD8+ T cells as described in the previous section could lead the tumor to be in a non-recognized state [8,34].

Cancerous cells are in a constant state of proliferation. Sometime during its growth, some parts of the tumor can be deprived of nutrition and oxygen due to imbalance between angiogenesis and tumor cells proliferation rate, thereby resulting in tumoral cell death. These tumoral cell deaths will trigger the uptake of tumor antigen by dendritic cells. The dendritic cells will further present the processed tumor antigen through MHC Class II molecules to various other immune cells, thereby triggering an enhanced immune response.

Nevertheless, in various cancers, the tumor cells expressed various cytokines, for instance, transforming growth factor beta (TGF-β), vascular endothelial growth factor (VEGF), interleukin-10 (IL-10), macrophage colony-stimulating factor (M-CSF), which blunt various functions of dendritic cells [35]. These anti-inflammatory cytokines act to prevent the maturation of dendritic cells until antagonizing the pro-inflammatory function of dendritic cells by promoting T regulatory cell differentiation [35]. Thereby, these molecules further prevent tumor recognition and T cell priming.

## 4. Immune Checkpoint Inhibition

Cancer cells with their tumor neo-antigens that are successfully recognized by T cells would hypothetically trigger a rejection process. The CD8+ T cells will be primed, then multiple cytokines and chemokines secretion will follow. There will be extensive recruitment of multiple pro-inflammatory cells toward the tumor microenvironment. Nevertheless, in many cancers, even with the presence of tumor cell specific antigens that have been successfully recognized by T cells, immune cells still fail to prevent tumor growth in many patients [36]. This phenomenon happens because some negative immuno-regulatory pathways exist to inhibit the function of CD8+ T cells in order to attack the tumor cells. This paradoxical coexistence of primed CD8+ T cells and the continued growth of tumors in cancer patients was known as the paradox of Hellström [37].

Multiple immune checkpoints molecules have been known to be able to inhibit the function of CD8+ T cells by sending negative regulatory signals. Some of these molecules are Cytotoxic T-lymphocyte antigen 4 (CTLA-4/ CD152), Programmed cell death protein 1 (PD-1/ CD279), Lymphocyte-activation gene 3 (LAG-3), T cell immunoglobulin and mucin domain 3 (TIM-3), and V-domain immunoglobulin suppressor of T cell activation (VISTA) [38,39,40]. Generally these immunosuppressive molecules work by exhausting the CD4+ and CD8+ T infiltrating lymphocytes; promoting T regulatory differentiation; abrogating the CD8+ cytotoxic effect; downregulating T cell proliferation; and all those effects culminating in enhancing immune tolerance [38,39].

With the accumulated understanding about the mechanism of immune tolerance mediated by those immune checkpoints molecules, today, many immune checkpoint inhibitors have been developed to reverse that mechanism. The most extensively studied immune checkpoints molecules are CTLA-4, PD-1, and Programmed cell death Ligand 1 (PD-L1). The PD-L1 is the ligand for PD-1 receptor. Both PD-1 and PD-L1 are expressed in broad range of immune cells and tumor cells including T cells, B cells, dendritic cells, and myeloid cells [41]. The interaction between PD-1/PD-L1 results in abrogation of fully primed effector CD8+ cytotoxic function [41]. Thus, by blocking PD-1 or PD-L1, we can expect a reversal of that immune-inhibitory effect. (Figure 2)

The CTLA4 checkpoint inhibitor is another important immune checkpoint molecule that exerts its effect during T cell priming. The anti CTLA-4 was thought to exert its immunotherapeutic effect by blocking interaction between the CTLA-4 receptor on T cells and B7 receptor on the antigen presenting cell [41]. The B7 receptor is a co-stimulatory molecule of the MHC-TCR complex. Upon binding of B7 to CD28 on the T cell, it initiates a signal that promotes T cell survival and proliferation. The binding of CTLA-4 with B7 would render CD28 unable to bind with B7, thus disabling the T cell priming process [41]. However, recent evidence indicating anti CTLA-4 in fact exerted its immunotherapeutic effect not by blocking B7/CTLA-4 interaction, but most likely because the anti CTLA-4 binding to T cells resulted in antibody-dependent cellular cytotoxicity (ADCC) [42]. The CTLA-4 blocking was attributed to decreased Treg levels, as these Treg cells showed elevated CTLA4 expression [42]. Therefore, that process results in the elimination of T reg cells with the positive effect of loss of major immunosuppressive cells within the tumor microenvironment (Figure 2).

## 5. Functional T Cells for Effective Immune Attack

Even though the administration of immune checkpoint inhibitors described in the previous section have successfully reversed the immunosuppressive mechanism by cancer cells in some cases, however, a substantial number of cases still did not show tumor regression or, at most, only a temporary tumor regression. This persistent immune tolerance is due to the proliferation of dysfunctional to exhaustive T cells within the tumor microenvironment [43,44,45]. An in-depth study analyzing the transcriptome of every single immune infiltrating lymphocytes in melanoma samples revealed that majority of CD8+ T cells were indeed lacking a complete cytotoxic gene expression, thus making them dysfunctional [43]. Furthermore, those dysfunctional CD8+ T cells were clonal and very proliferative within the tumor microenvironment [43] (Figure 3).

Apart from the presence of highly proliferative dysfunctional CD8+ T cells, tumor rejection is also hampered by the presence of a highly immunosuppressive microenvironment elicited by T regulatory cells. T regulatory cells that exhibit an immunosuppressive phenotype generally express CD45RA-, FOXP3+high, CD4+ and CD25+high markers [46]. A study revealed that those T regulatory cells were found to be abundant in the tumor sample compared to systemic circulation (10–50% vs. 2–5%) [46,47]. Various chemokines and cytokines secreted by tumor cells are thought to be the main perpetrator of T regulatory cells recruitment [46]. The T regulatory cells also express immune checkpoint molecules such as CTLA-4 and PD-1 [46,48]. The administration of anti CTLA-4 monoclonal antibody resulted in ADCC, thus reducing the number of T regulatory cells [42,46], while the anti PD-1 effect on T regulatory was not very clear. Some evidence suggests that anti PD-1, contrary to that expected, promotes an enhanced immunosuppressive T regulatory activity [46,48].

In order to enable a fully functioning immune system to reject tumor cells, the ultimate goal of functioning specific cytotoxic CD8+ T cells have to be achieved. Various strategies have to be developed to deal with the immune-tolerance mechanism elicited by tumor cells. A focus on the tumor microenvironment by detecting what went wrong on that particular cancer seems to be able to pave the way for a more comprehensive understanding of immune escape. Finally, it has to be understood that an effective immunotherapy is unlikely without tackling all aspects of immune evasion mechanism by the cancer cells. The following section will discuss the specific failure of each form of immunotherapy available.

## 6. Hurdles of Immunotherapy

There are various forms of immunotherapy available as cancer treatment. The oldest form of immunotherapy is in the form of pro-inflammatory cytokine administration such as interferon. Later, immune checkpoint inhibitors were developed based on monoclonal antibody. These immune checkpoint inhibitors are the most commonly utilized immunotherapy today. The other forms of immunotherapy are adoptive cell transfer and oncolytic virus vaccine [49,50]. Adoptive cell transfer is based on isolation or engineering of immune cells such as cytotoxic T cells or NK cells that are able to recognize tumor neo antigens [49,50]. Then, these adoptive cells are expanded ex vivo and then infused into the patients [49]. It is expected that those infused killer cells would recognize and attack the cancer cells. Oncolytic virus vaccine is based on engineered virus with its virulence gene deleted [51]. The administration of oncolytic virus into the tumor will trigger tumor infection, which later leads to cell lysis with positive inflammatory response [51].

The effectiveness of all modes of immunotherapy is very dependent on the intrinsic and dynamic tumor microenvironment of each patient. Any immunotherapy will only be effective if strong response of tumor recognition and tumor attack are elicited. In cytokine based immunotherapy, the interferon increases dendritic cell maturation, favors T helper 1 differentiation, increases cytotoxic function of Natural Killer (NK) cells, and increases tumor MHC class 1 expression [52,53]. However, interferon has also been shown to induce upregulation of immune checkpoints such as PD-L1 [54,55]. Therefore, immune tumor attack becomes suboptimal. Combining interferon and immune checkpoint inhibitor has a sound rationale. Phase I and II trials have shown the safety and initial efficacy of these combinations [56,57] Nevertheless, complete response was only observed in a small subset of patients [56,57]. This observation underscores that some other mechanisms were in place that rendered either tumor recognition or tumor attack suboptimal. 

Various immune checkpoint inhibitors have been approved and used in clinic. A phase III trial comparing chemotherapy and a single agent pembrolizumab, an anti PD-1, has shown a remarkable survival benefit of pembrolizumab over chemotherapy in metastatic non-small cell lung cancer [58]. Furthermore, an analysis of a KEYNOTE-001 study showed that 16% of patients receiving pembrolizumab for metastatic melanoma had a durable complete response [59]. These significant findings were clinically very meaningful, but unfortunately was only observed in small subsets of patients. Immune checkpoint inhibitors generally act to re-invigorate the tumor immune attack, as discussed in the previous section. In a condition where there is a lack of tumor immune recognition per se, then immune checkpoint inhibitors are unlikely to be clinically beneficial.

Adoptive cell transfers can provide superb clinical outcomes in some circumstances. In B cell acute lymphoblastic leukemia, genetically engineered chimeric antigen receptors (CARs) T cells resulted in up to 80% complete remission [60,61]. However, clinical success of adoptive cell transfer in other types of cancers is limited. In adoptive cell transfer, it is a prerequisite to identify the target of immune attack, which is the tumor expressed antigens [62]. Most solid tumors do not have exclusive tumor antigens. Most tumor expressed antigens were also expressed in normal tissue, though in lower concentration. The targeting of non-exclusive antigens could result in severe toxicity [60,63]. Therefore, the main hurdle of adoptive cell transfer is identifying exclusive tumor neo antigens that are suitable to become the target of immune attack.

Augmenting the function of natural killer cells or adoptive transfer of NK cells have shown some clinical benefit as immunotherapy [64,65,66,67]. Unlike T cells, NK cells do not depend on MHC Class I expression to recognize tumor cells [67]. There are numerous activating and inhibiting receptors on NK cells that constantly interact with other cells [64]. In normal cells, the ligands for NK cell activating receptors are poorly expressed, thereby maintaining NK cells in a non-activated state [64]. In tumors expressing inhibitory receptors, the ligand of NK cells will blunt the activity of NK cells. Furthermore, an immunosuppressive tumor microenvironment will also result in exhausted NK cell phenotype [64], similar to exhaustive T cells. Therefore, a complete understanding of tumor, immune, and its tumor microenvironment interaction is necessary to tackle the hurdle of NK cells immunotherapy.

Oncolytic virus is another interesting type of immunotherapy. Talimogene laherparepvec (T-VEC), this Food and Drug Administration (FDA) approved oncolytic virus, has been tested in a phase III trial that showed superior clinical benefit in patients with advanced melanoma compared to cytokine granulocyte-macrophage colony-stimulating factor injection [68]. Though oncolytic virus showed a promising result, the development is very challenging. There are various hurdles of effective oncolytic virus including the rapid elimination of virus by the host immune system in a patient with pre-existing viral antibody, the inadequate specificity of virus infectivity toward tumor cells, until the pre-existing immunosuppressive tumor microenvironment [69,70]. The latter would eventually render an inadequate tumor immunity.

The hurdles of immunotherapy have to be viewed in a more comprehensive perspective. The hurdles are due to the establishment of inherent subversion mechanisms of immune recognition and immune attack of tumor cells. Immunotherapy alone in most cases would not be able to counteract all of these mechanisms. One way to counteract these mechanisms is by combination therapy. Various commonly employed anti-cancer treatments are able to tackle some of these mechanisms. When combined with immunotherapy, multiple mechanisms could be reversed simultaneously and result in better tumor control. Next, in the following section, we will discuss various commonly employed anti-cancer treatment and its relationship with tumor immunity. Later in this review, we propose a way to rationally combine these treatments.

## 7. Hot Tumor Induction by Radiation Therapy

Cold and hot tumors is a notion for tumors with heavy tumoral T cell infiltration or scarce tumoral T cell infiltration, respectively [36]. Cold or hot tumors are dependent upon the specific tumor intrinsic mechanism in escaping immune recognition and attack. A hot tumor generally escapes tumor attack, but they are readily recognized by immune cells, while cold tumor could escape both immune recognition and immune attack. Therefore, a hot tumor confers a better prognosis than a cold tumor [71,72]. There is still a spectrum of tumor in between hot and cold tumors, known as altered tumors. In order to improve the treatment result, conversion to a more favorable hot tumor is a more detailed discussion of various kinds of tumor states based on their immune signatures.

Radiation therapy is a traditional modality that has been used for many decades for cancer treatment. In the past, radiation therapy was generally considered immunosuppressive. Lymphocytes are very sensitive to radiation [73]. A radiation dose of just 2 Gy to the whole body could result in pancytopenia [74]. Radiation also used to be indicated to treat benign conditions due to inflammation such as osteoarthritis. A low dose of 0.5 Gy per fraction with a total dose of 6 Gy has been historically used to treat inflammation conditions with great success [75]. However, numerous evidence indicates that radiation can also be immuno-stimulatory when given at a higher dose in certain conditions [76,77,78,79,80]. 

Radiation can enhance inflammation response by increasing the production of interferon type I (IFN-1) [80,81]. IFN-1 will later promote recruitment of dendritic cells and increase antigen presentation. Radiation results in double strand DNA (dsDNA) breaks. The broken DNA fragments within the cytoplasm of the cells will trigger the activity of the cyclic GMP-AMP synthase (cGAS) [82]. The activity of cGAS will eventually lead to the activation of stimulator of IFN genes (STING) [81,83]. The STING will phosphorylate interferon regulatory factor 3 (IRF3), resulting in activation and translocation of IRF3 into the nucleus. This IRF3 acts as transcriptor activator of the IFN-1 gene [81,83]. All these processes eventually lead to the transcription of the IFN-1 gene.

Nevertheless, not all radiation will result in stimulation of IFN-1. A very high dose of radiation, beyond 20 Gy in a single fraction was not associated with the increase in IFN-1 production [80]. This was due to the elimination of DNA fragments in the cytoplasm by DNA exonuclease Trex1 [84]. A higher radiation dose was associated with higher Trex1 expression, typically occurring in a dose of 12 Gy and beyond, depending on cell types [81]. The high expression of Trex1 in cytoplasm resulted in elimination of the DNA fragment necessary to induce the cGAS-STING-IFN1 pathway [80,81]. Meanwhile, an 8 Gy radiation dose, a moderately high dose of radiation compared to the conventional radiation of 2 Gy, has been shown to produce a greater number of DNA fragments in the cytoplasm of the irradiated cells [80], thus this higher cytoplasmic DNA enables a greater stimulation of IFN-1.

Radiation therapy was also able to increase the recognition of tumor cells by the immune system through upregulation of MHC Class I molecules in tumor cells [76,77,85,86]. Induction of MHC Class I expression after radiation therapy was related with IFN-1, in particular, IFN-β secretion by tumor cells themselves [86]. The induction of MHC Class I expression was due to stimulation of IFN receptors by IFN-β in an autocrine and paracrine fashion [86]. However, the upregulation of MHC Class I after radiation could also occur without IFN-β. Radiation has been proven to increase the expression of NLRC5 [85], an MHC class I transactivator, as discussed above. The induction of NLRC5 following radiation induced MHC class I expression independent of IFN-β [85]. The higher MHC Class I expression will later lead to improved tumor recognition through enhanced antigen presentation.

Irradiated tumor cells would also undergo immunogenic cell death with the release of tumor antigens and various danger-associated molecular patterns (DAMPs) molecules [87,88]. This immunogenic cell death will further increase the process of immune recognition by antigen presenting cells, as discussed above. The secretion of chemo attractants such as CXCL16 was also shown to be enhanced after radiation therapy [89]. All these factors contribute to enhanced recruitment of CD8+ T cells into tumor microenvironment and overcome at least part of the tumor resistance of immune recognition. When combined with immune checkpoint inhibitors, theoretically, part of both resistance of tumor recognition and tumor attack would be overcome.

Though radiation therapy has shown promising results in creating a so called in situ tumor vaccine that proved to be beneficial when combined with immune checkpoint inhibitor [36] a greater understanding is necessary. As radiation can be immune-stimulatory and at the same time immunosuppressive, greater fundamental studies on the radiation dose, mode of the delivery, radiation target, number of radiation fractions, and how to sequence the combination with immune checkpoint inhibitor are required. However, for now, radiation therapy has proven to have a plausible rationale in tackling some resistance of immunotherapy.

## 8. Synergistic Effect of Chemotherapy and Immunotherapy 

Chemotherapy has been used for decades to treat cancer. Chemotherapy induces cell death through direct cytotoxicity by various mechanisms include inhibition of cell proliferation. However, a sustained tumor regression after chemotherapy is now known to be dependent on the immune system [90,91]. In an animal model, chemotherapy combined with a macrophage depleting agent resulted in tumor regression only for a short period of time, while a more sustained regression was found without a macrophage depleting agent [90]. This finding indicated that even with traditional treatment such as chemotherapy, immune system plays a great role in determining the success of that particular treatment.

Chemotherapy has been shown to be able to synergize with immunotherapy to improve cancer killing [92,93,94]. Chemotherapy agents that induce immunogenic cell death such as anthracyclines (i.e., doxorubicin), platinum compounds (i.e., oxaliplatin), and alkylating agents (i.e., cyclophosphamide) are generally more likely to synergize with immunotherapy [93,94,95]. Chemotherapy induced immunogenic cell death requires induction of stress signals or DAMPs. The stress signals have to be either translocated to plasma membrane in order to be recognized by the dendritic cells or secreted by the tumor cells. These stress signals include calreticulin (CALR), protein disulfide isomerase family A member 3 (PDIA3/ERp57), heat shock protein (HSP70/HSP90), ATP, HMGB1, and type I IFN [94,96].

The secretion or exposure of stress signals is the prerequisite of chemotherapy induced immunogenic cell death. The initial event is marked by stress toward endoplasmic reticulum. Then, an activation of unfolded protein response (UPR) is initiated with various downstream effects [96]. One of the effects is the increased synthesis of endoplasmic reticulum chaperones [96]. These endoplasmic reticulum chaperones will later be phosphorylated by eukaryotic translation initiation factor 2A (eIF2A), resulting in translocation and exposure of stress signals [94]. Some chemotherapy agents such as cisplatin are unable to initiate the endoplasmic reticulum stress response, thus inherently unable to generate immunogenic cell death [94].

Other important stress signals (ATP, HMGB1, and type I IFN) released by tumor cells are also important in triggering immunogenic cell death. When chemotherapy induces endoplasmic reticulum stress, the process of autophagy proceeds with the accumulation of ATP in autolysosomes [96,97]. Then, an exocytosis process occurs with the result of ATP secretion [97]. The HMGB1 has also been shown to induce the process of immunogenic cell death, especially when present in reduced state [96]. Though HMGB1 can also reverse the process of pro-inflammation when present in the oxidative state, it has been shown to be a critical molecule in chemotherapy induced immunogenic cell death [96]. Type I IFN has been known to be a potent inflammatory chemokine [95]. The secretion of type I IFN occurring following chemotherapy has been shown to be dependent on Toll-like receptors 3 (TLR3) [96,98]. The activation of TLR3 further leads to the secretion of type I IFN to the tumor microenvironment, thus recruiting immune cells [98].

Apart from secreting pro inflammatory stress signals, there are also other mechanisms of chemotherapy promoting inflammation. Some chemotherapy such as cyclophosphamide, paclitaxel, and temozolamide could deplete Treg cells [92,93]. Furthermore, paclitaxel has been shown to selectively induce greater cell death in myeloid-derived suppressor cells (MDSCs) compared to dendritic cells [99]. These MDSC cells are potent anti-inflammatory immune cells. The differential sensitivity to paclitaxel was thought to be due to a greater abundance of P450 reductase enzyme in MDSC cells compared to dendritic cells [99]. The P450 reductase enzyme metabolizes this paclitaxel into its toxic active metabolite, thus selectively killing more MDSC cells and eliminating negative regulatory cells [92,99]. Nucleoside analogs such as gemcitabine and 5-fluorouracil could also reduce the number of MDSC cells [92]. Apart from that, gemcitabine has been shown to significantly increase MHC Class I expression in tumor cells [100], therefore increasing the chance of tumor immune recognition.

Combining chemotherapy with immunotherapy in an attempt to increase tumor killing without resulting in excessive side effects is possible, though challenging. As described above, chemotherapy could induce immunogenic cell death, favoring increased tumor recognition until tumor attack. However, multiple mechanisms elicited by tumor cells impede the process of tumor cells killed by immune cells. An in-depth understanding of immune response and immune contexture of each associated chemotherapy agent is necessary to properly combine both modalities. The sequence of combination, type of chemotherapy agent combination, dosage, timing, number of cycles, and response assessment are all important clinical questions yet to be explored in order to obtain the best synergy between chemotherapy and immunotherapy.

## 9. Targeted Therapy Reverses Immunotherapy Tolerance

The understanding of the molecular carcinogenesis process has led to the development of various targeted agents. Specifically, these agents were initially developed to put a brake or even reverse the derailed intracellular signaling processes that lead to the development of cancer. However, these targeted agents have also been shown to be able to modulate the immune system to enhance immune recognition and immune attack in some instances [92]. Therefore, there is a rationale to combine targeted therapy with immunotherapy to enhance immune mediated tumor killing. The mechanisms of these immune related tumor killing is discussed below.

Many types of targeted therapies are based on monoclonal antibody functioning to inhibit the abnormal function of a specific receptor on the plasma membrane (i.e., Anti-Epidermal Growth Factor Receptor (EGFR)). An anti EGFR, for example, cetuximab has been shown to induce internalization of the EGFR, followed by degradation of EGFR through cellular lysosomes [101]. Thereby, the process results in downregulation of the EGFR and suppression of proliferation. Apart from that, these monoclonal antibody targeted agents can also induce immune related cell killing through the process of ADCC and complement dependent cytotoxicity (CDC) [101]. However, as EGFR is also expressed in normal cells [102], enhancing the effect of ADCC or CDC of these monoclonal antibodies could potentially increase the toxicity. Thus, this limits the exploitation of immune mediated cell killing using monoclonal antibodies targeted agents.

Another available targeted therapy is based on small molecules that inhibit the intracellular tyrosine kinases. For instance, EGFR has an intracellular tyrosine kinase domain. The inhibition of abnormally activated EGFR will suppress the proliferation of tumor cells. Furthermore, the inhibition of EGFR with anti-EGFR tyrosine kinase inhibitor (TKI) (i.e., Afatinib) has been shown to upregulate the MHC Class I receptor [36,103]. The abnormal activation of EGFR due to various kinds of mutation within the cancer resulted in sustained downstream activation of mitogen-activated protein kinases (MAPK) pathway [103]. The activation of the MAPK pathway was associated with suppression of the MHC Class I expression [36,103]. Therefore, the inhibition of EGFR with TKI or any intracellular kinases involved in the MAPK pathway resulted in enhanced upregulation of the MHC Class I molecule. The increase in the MHC Class I receptor would enhance the antigenicity of the tumor cells and aid tumor recognition.

Another important molecule that drive cancer progression is vascular endothelial growth factor (VEGF). Currently, there are various TKI targeted agents that target VEGF, for instance, sunitinib. VEGF has been known for some time to be an important molecule in cancer progression. VEGF can induce angiogenesis, increase vascular permeability, remodel extracellular matrix, and sustain self-renewal until recruiting T regulatory cells to the tumor microenvironment [104]. Therefore, blocking VEGF with anti VEGF is mechanistically rational. Apart from inhibiting T regulatory cell recruitment to the tumor microenvironment [92], anti VEGF TKI has also been shown to be able to reverse the immune-inhibitory environment created by the tumor. 

As discussed above, one of the hurdles of effective tumoral immune attack is due to dysfunctional cytotoxic T cells. The VEGF molecule is one of the molecules responsible for causing the stimulation of various immune-inhibitory molecules such as PD-1, TIM-3, and LAG-3 in cytotoxic T cells, rendering the immune cells dysfunctional [92,105]. The inhibition of VEGF resulted in reduced expression of the immune-inhibitory molecules in cytotoxic T cells, thus reversing the T cell exhaustion within the tumor microenvironment [105]. Furthermore, combining anti VEGF with anti PD-1 has been shown to enhance tumor shrinkage [105], validating the concept of combining those two modalities.

## 10. Treatment Based on Cancer Immune Landscape

As discussed above, there are many mechanisms in which the tumor can put forward to escape host immune surveillance and attack. However, there is a common agreement that the immune system plays a tremendous role in harnessing cancer growth and even dictating the chance of relapse after treatment [106]. Furthermore, even with the conventional cancer treatment modalities such as radiotherapy and chemotherapy, they also rely very much on the host immune system to sustain tumor regression. A cancer immune profile, therefore, would significantly determine the prognosis of a particular patient [36]. Moreover, a sound cancer treatment strategy can be devised by utilizing the information on the tumor microenvironment immune contexture. It is expected that knowing a precise mechanism of tumor immune escape will allow treatment administered with a greater chance of success (Figure 4). 

In the following discussion, we proposed and summarized a combinatorial strategy that can be utilized to improve tumor recognition and tumor attack by knowing the exact cancer immune landscape. The proposed combinatorial strategy is limited to combination of various immune checkpoint inhibitors with other traditional agents that have been described above. There are many novel treatments such as adoptive cell transfer, cancer vaccine with oncolytic virus, NK cell treatment, and so on, which can also be used in selected settings as cancer immunotherapy [107,108,109]. However, these novel treatments are not readily available in most cancer centers. Therefore, this proposed combinatorial strategy can be adopted more readily and widely. 

The immune profile in a tumor microenvironment of a particular cancer can be classified into hot tumors (inflamed tumor), altered tumors, and cold tumors (non-inflamed tumor) [36,110]. An altered tumor is a wide spectrum in between inflamed and non-inflamed tumor. For the purpose of simplification, altered tumors are generally classified into immunosuppressed and excluded states [36]. An immunosuppressed tumor is a state in which there are tumor infiltrating lymphocytes (sometimes possibly with immune-inhibitory phenotype) intermingled between tumors. There are also relatively abundant negative regulatory cells such as T regulatory cells or MDSCs between tumor cells in immunosuppressive tumors, while in excluded tumor, there are abundant lymphocytes at the periphery or stromal border of the tumors. In this excluded tumor, there is a kind of barrier for effective tumor infiltration.

In a hot tumor, immune system is generally able to recognize the tumor cells. It is marked by a dense infiltration of tumor infiltrating lymphocytes within tumor microenvironment. Nevertheless, the expression of various immune-inhibitory molecules by tumor cells such as PD-1, LAG-3, TIM-3, and VISTA results in abrogation of tumor attack by cancer cells. In such cases, the use of immune checkpoint inhibitors is theoretically sufficient to promote immune attack and result in tumor regression. In immunosuppressed tumors, this state closely resembles hot tumors, however, the presence of multiple negative regulatory cells within the tumor microenvironment resulted in a more complex failure of tumor immune attack. This immunosuppressed tumor can be counteracted by combining immune checkpoint inhibitors with regulatory cell depleting agents [111]. 

Another altered state of tumor, the excluded tumor, less resembles hot tumors. This state of excluded tumor is probably, to some extent, still able to be recognized by tumor cells, determined by abundant lymphocytes on the tumor periphery, but not intermingled between tumor cells. The reason for this is probably a lack of various chemokines or cytokines necessary to recruit these lymphocytes to the tumor microenvironment [110]. Furthermore, tumor cells could also affect the extracellular matrix and local vasculature, for instance, by secreting VEGF. The presence of VEGF does not favor the process of T cell recruitment to the tumor microenvironment [36]. This excluded tumor can be initially counteracted by stimulating the production of various pro-inflammatory chemokines and administration of an anti-angiogenesis agent. 

Once this excluded tumor receives an initial treatment and become more infiltrated by lymphocytes, it is required to have a second look at its tumor microenvironment. It is possible that other mechanisms are in place to hinder tumor attack. A study assessing the tumor microenvironment of primary and relapsed tumor found that the tumor microenvironments were completely different [112]. The relapsed tumor was bathed with numerous immunosuppressive cells [112]. Furthermore, mutational signatures of a clinical cancer specimen before treatment and the specimen during recurrence after treatment showed a complete different mutational signature [113]. This dynamic tumor mutation might also contribute to different stimuli and result in a different tumor microenvironment. These findings indicated that the tumor microenvironment is certainly dynamic and constantly evolving [112,114]. Therefore, after an initial treatment that would disrupt its microenvironment, a second assessment of the tumor microenvironment is necessary. Other treatments, for instance, immune checkpoint inhibitors have to be added later on, depending on this changing tumor immune landscape.

A cold tumor has none to very scarce tumor infiltrating lymphocytes within the tumor microenvironment. This cold tumor phenotype indicates that the tumor is not recognized by immune cells. One way to counteract this cold tumor is by inducing tumor recognition by various means such as upregulating tumor MHC Class I receptors, stimulating tumor neo-antigens until enhancing the recognition and activation of dendritic cells. The induction of tumor recognition can be achieved by administering radiotherapy in a certain fashion where radiotherapy is expected to work as a kind of tumor vaccine. Apart from radiotherapy, the administration of agents that can elicit immunogenic cell death is also an alternative, as previously discussed. However, even though the resistance of tumor recognition has been solved, there is still a possibility of various other mechanisms elicited by tumor cells that are in place that fail the tumor immune attack [114]. Similar to the case of excluded tumors, in this kind of cold tumor, a staged approach for a second assessment is required. A second biopsy will determine the tumor immune profile after successful induction of tumor recognition. The information on further evolution of the tumor immune landscape is again used to guide further choice of treatment. 

Though, theoretically, this approach seems to be promising, there are various hurdles in implementing this approach in clinical trials and clinical settings. The characterization of the tumor immune landscape is technically complicated and challenging. Furthermore, there is a lack of standardized guidance on the steps and techniques required for immune landscape characterization. Most clinical trials combining immunotherapy with radiotherapy, chemotherapy, or even targeted therapy are not selecting patients treatment based on cancer immune landscape characterization [115,116,117,118]. However, biological cancer specimens are generally stored for further retrospective analysis to understand the effect of the cancer immune landscape and various combinatorial treatment. Future clinical trials with specific emphasis on selecting treatment strategy based on cancer immune landscape compared with just standard practice is required to prove the benefit of this approach.

## 11. Conclusions

Immune escape by cancer is mainly due to the failure of recognition and failure of attack. There are abundant mechanisms in which cancer cells can escape immune recognition including downregulating MHC Class I and promoting immune suppressive CD4+ T regulatory cells/MDSCs within the tumor microenvironment. The failure of immune attack could occur due to upregulation of various immune checkpoint molecules, thus making the immune cells dysfunctional and exhaustive. Failure in cancer attack can be reversed partly by administering immune checkpoint inhibitors, however, abundant proliferating dysfunctional T cells within the tumor microenvironment render those immune checkpoint inhibitors useless. Therefore, it is necessary to understand the basic mechanism that fail immune surveillance.

There are many combinatorial strategies that can be used to reverse the resistance of immunotherapy to synergize with immunotherapy or even to sensitize the effect of immunotherapy. Many traditional modalities of cancer treatment can be combined with immunotherapy to improve cancer killing. However, a prior characterization of the cancer immune landscape seems to be necessary due to the heterogenicity of the tumor microenvironment, even in the same kind of tumor in the same patient, but in a different period. This information is necessary in order to understand the mechanisms that underlie the failure of immune recognition and immune attack. Then, the best strategy to target those mechanisms of immune escape can be developed. The future of cancer treatment is going to evolve very rapidly. Hopefully, with the understanding of this basic mechanism of immune escape, further trials can be designed utilizing this strategy with the main aim to increase our ability to fight and control cancer.

## Figures and Tables

**Figure 1 molecules-25-04096-f001:**
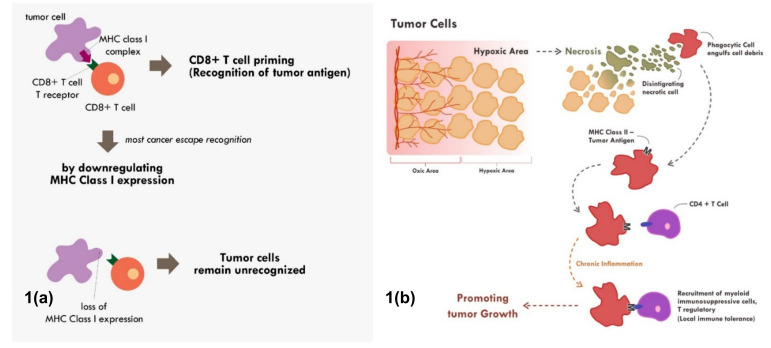
Mechanisms of tumor escape immune recognition. (**a**) Through downregulation of MHC Class I in tumor cells, rendering tumor cells unrecognized. (**b**) Through engulfment of tumor debris and further presentation through MHC Class II and APC cells (chronic process of that event leads to chronic inflammation resulting in sensitization and differentiation of CD4+ T cells toward immune suppressive CD4+ T regulatory cells phenotype).

**Figure 2 molecules-25-04096-f002:**
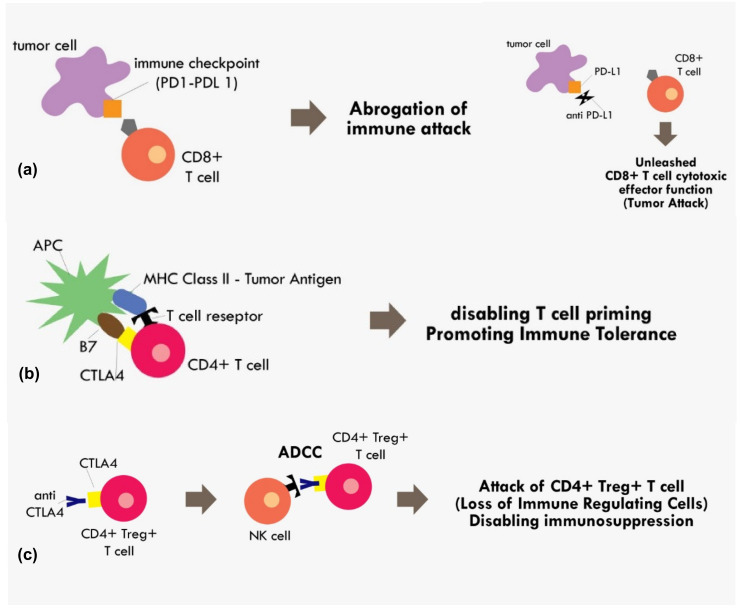
Mechanism of immune checkpoint inhibitions (**a**) PD1–PDL1 binding results in abrogation of those primed CD8+ T cells, while blocking the PD1–PDL1 axis results in unopposed CD8+ effector T cells functions. (**b**) Binding of costimulatory CTLA4 in CD4+ T cells with B7 in APC in the MHC Class II-Tumor antigen presentation process results in cancellation of T cell recognition and priming (**c**) binding of CTLA4 receptor in CD4+ Treg+ T cell by monoclonal antibody results in the antibody dependent cellular cytotoxicity (ADCC) process, thus eliminating immunosuppressive T regulatory cells within the tumor microenvironment.

**Figure 3 molecules-25-04096-f003:**
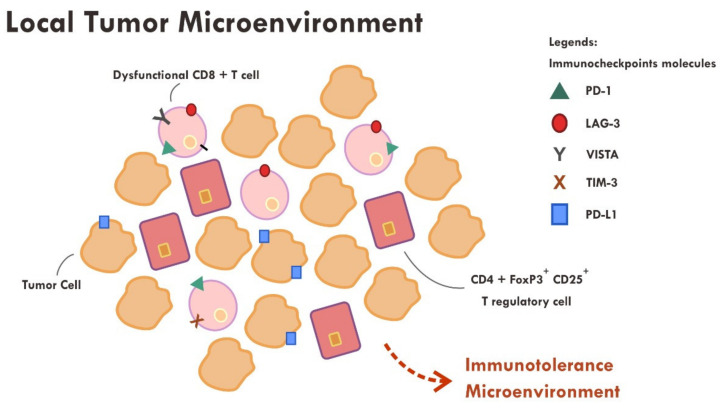
Abundance of dysfunctional CD8+ T cells expressing various immune inhibitory molecules (PD-1, LAG-3, VISTA, TIM-3, PD-L1) within the tumor microenvironment resulted in failure of tumor attack.

**Figure 4 molecules-25-04096-f004:**
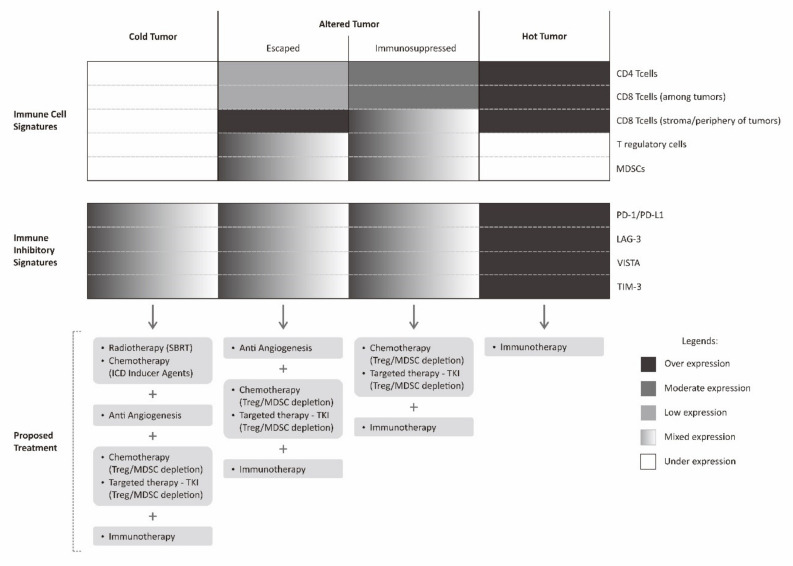
Proposed multimodality treatment combining immunotherapy with various other treatments based on cancer immune landscape. Immunotherapy described in this figure is limited to checkpoint inhibitor.

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
