# Peer review of "Tackling Resistance to Cancer Immunotherapy: What Do We Know?"

_molecules, 2020, doi:10.3390/molecules25184096_

Round 1

Reviewer 1 Report

In their review article, Ghondhowiardjo et al. set out to highlight major obstacles preventing the widespread applicability of immunotherapy, and propose their own rationale for how combinatorial therapies can subvert these obstacles. Although the authors do a sufficient job in describing their main points, this article could benefit from further in-depth description of each of the modes of immunotherapy. Specifically, in terms of the unique challenges that face each therapeutic modality independently.

In addition, the authors should take care to address the following:

  1. The entire review should be carefully proofread for correct English usage, as there are multiple instances of incomplete sentences, incorrect sentence structure, and subject-verb mismatch. It would be highly recommended to have a native-English speaker review and revise the document if possible.
  2. The introduction spends some time introducing readers to human immunology and the principles of immune recognition. This does not contribute to the main narrative of the manuscript, and is a very cursory description of complicated principles. This manuscript would benefit from a removal of these sections.
  3. The authors specify that the goal of this review is to highlight how (1) recognition and (2) attack are two of the major hurdles facing immunotherapy. As such, it would strongly support their premise if they could provide tangible evidence for both a recognition-based hurdle and an attack-based hurdle facing a given form of immunotherapy. (ie, What is a recognition-based hurdle for antibody therapy? For cell-based immunotherapy? For vaccine-therapies?). Furthermore, although the abstract dictates that the goal of this manuscript is to detail the challenges facing immunotherapy, it instead chooses to focus on forms of cancer therapy, not the challenges therein.
  4. Page 4 onwards contains a thorough description of T priming and recruitment to the tumor microenvironments – the authors should take care to detail the incidence of other anti-tumor immune cells (such as NK cells), which can also significantly contribute to overall anti-tumor immunity.
  5. Lastly, the authors do not propose their own solution to these problems, as is stated in the abstract.

Author Response

In their review article, Ghondhowiardjo et al. set out to highlight major obstacles preventing the widespread applicability of immunotherapy, and propose their own rationale for how combinatorial therapies can subvert these obstacles. Although the authors do a sufficient job in describing their main points, this article could benefit from further in-depth description of each of the modes of immunotherapy. Specifically, in terms of the unique challenges that face each therapeutic modality independently.

Thank you very much for your suggestion. We have added a section describing each mode of immunotherapy. The section added was section 6 “Hurdles of Immunotherapy”. This section specifically described the mechanism of every mode of immunotherapy with its possible cause of failures and challenges.

In addition, the authors should take care to address the following:

1. The entire review should be carefully proofread for correct English usage, as there are multiple instances of incomplete sentences, incorrect sentence structure, and subject-verb mismatch. It would be highly recommended to have a native-English speaker review and revise the document if possible.

We have made a revision on the language. Yes, we identified multiple grammatical errors, but we have revised it accordingly.

2. The introduction spends some time introducing readers to human immunology and the principles of immune recognition. This does not contribute to the main narrative of the manuscript, and is a very cursory description of complicated principles. This manuscript would benefit from a removal of these sections.

Okay, initially we thought it was necessary to introduce reader on quick basic immunology. However, since it was thought to make it more complicated, we agreed and we have omitted that section on the introduction.

3. The authors specify that the goal of this review is to highlight how (1) recognition and (2) attack are two of the major hurdles facing immunotherapy. As such, it would strongly support their premise if they could provide tangible evidence for both a recognition-based hurdle and an attack-based hurdle facing a given form of immunotherapy. (ie, What is a recognition-based hurdle for antibody therapy? For cell-based immunotherapy? For vaccine-therapies?). Furthermore, although the abstract dictates that the goal of this manuscript is to detail the challenges facing immunotherapy, it instead chooses to focus on forms of cancer therapy, not the challenges therein.

We have added section 6 which outlined specifically the hurdles of each mode of immunotherapy. Yes, it is true that basic failure of cancer treatment from immune perspectives are mainly due to failure of recognition and attack, however, we cannot simply point out whether it is a recognition or an attack failure in each mode of immunotherapy. The inherent mechanisms of immune escape are the main contributors of cancer treatment failure. Therefore, in our first few sections, we elaborated extensively the possible processes of those inherent mechanism that result in immune escape, then the hurdles of immunotherapy come in. They are discussed in view of the general mechanism of immune escape to give readers a broad perspective.

4. Page 4 onwards contains a thorough description of T priming and recruitment to the tumor microenvironments – the authors should take care to detail the incidence of other anti-tumor immune cells (such as NK cells), which can also significantly contribute to overall anti-tumor immunity.

It is true that there were increasing evidence that support the role of NK cells as immunotherapy. Though, there are still a lot of unknown in NK cells cancer therapy. We have added the NK cell therapy in part of section 6.

5. Lastly, the authors do not propose their own solution to these problems, as is stated in the abstract.

We actually did propose a solution. Section 10 in this revised manuscript detailed the proposed strategy. The figure 4 summarized clearly the proposed treatment strategy based on the immune landscape. Though, this strategy still required a lot of testing in the context of clinical trials to fine tune it. We have also stated that in the manuscript.

Reviewer 2 Report

This is a well structured comprehensive review of the mechanisms of action and those of failure of immunotherapy in cancer. The synergistic effect of immunotherapy with radiotherapy as well as with chemotherapy is also adequately supported with preclinical evidence. The role of targeted therapy in modulating appropriate immune response is discussed with some pathway-specific in vitro examples and with a reference to one clinical study. Unfortunately, clinical data in support of the title is scanty. References #93-97 underline the benefit of immunotherapy in addition to standard treatment of colorectal, breast, and lung cancers, but do not give much guidance as to how one can tackle resistance to cancer immunotherapy in daily practice. I guess prospective readers of this review will also miss those suggestions.

Beyond the exploration of the immunological landscape in cases of failing immunotherapy I would be curious to see how the molecular genetic profile of specific tumors changes not only during the development of primary immunotolerance but also during the development of a secondary immunotolerance which presents as a failure of immunotherapy. It is highly likely that these data will come from studies on clinical samples of patients whose disease turns into a progression following remission during checkpoint inhibitor treatment.

Author Response

This is a well structured comprehensive review of the mechanisms of action and those of failure of immunotherapy in cancer. The synergistic effect of immunotherapy with radiotherapy as well as with chemotherapy is also adequately supported with preclinical evidence. The role of targeted therapy in modulating appropriate immune response is discussed with some pathway-specific in vitro examples and with a reference to one clinical study. Unfortunately, clinical data in support of the title is scanty. References #93-97 underline the benefit of immunotherapy in addition to standard treatment of colorectal, breast, and lung cancers, but do not give much guidance as to how one can tackle resistance to cancer immunotherapy in daily practice. I guess prospective readers of this review will also miss those suggestions.

Thank you very much for your suggestion and comment. We have added section 6 on the revised manuscript. This section detailed each mode of immunotherapy together with its clinical evidence. In the context of tackling resistance to cancer immunotherapy, the manuscript has to be viewed in the entirety. We described in details the problem of inherent mechanism of immune escape. Tackling resistance to cancer immunotherapy, in our view, cannot be solved by judging the mechanism of immunotherapy alone. Though, the mechanism of immunotherapy is important, but since there are a lot of factors come and interplay together that result in failure of cancer treatment, therefore, tackling resistance to cancer treatment failure need a comprehensive view too. We also put forward a strategy to improve cancer treatment by combining immunotherapy and various other agents. May be, that can be viewed as our tangible effort describing method in tacking resistance to immunotherapy.

Beyond the exploration of the immunological landscape in cases of failing immunotherapy I would be curious to see how the molecular genetic profile of specific tumors changes not only during the development of primary immunotolerance but also during the development of a secondary immunotolerance which presents as a failure of immunotherapy. It is highly likely that these data will come from studies on clinical samples of patients whose disease turns into a progression following remission during checkpoint inhibitor treatment.

Yes, thank you very much for pointing that out. This is highly relevant. Tumor microenvironment is dynamic and constantly evolving. Though the data on this issue is scant, but we have managed to add and point this out in section 10 in our revised manuscript. We pointed out that tumor microenvironment on primary tumor and relapsed tumor could be completely different, thus underpinning the importance of second assessment after initial treatment.

Round 2

Reviewer 2 Report

Well done. All criticisms have been thoroughly solved and appropriate supplementation was performed to achieve full comprehensiveness.

This manuscript is a resubmission of an earlier submission. The following is a list of the peer review reports and author responses from that submission.